# Synthesis of Iron Gallate (FeGa₂O₄) Nanoparticles by Mechanochemical Method

**Musa Mutlu Can** [1,*] **, Yeşim Akbaba** [1] **and Satoru Kaneko** [2,3,4]

1   Renewable Energy and Oxide Hybrid Systems Laboratory, Department of Physics, Faculty of Science, Istanbul University, Vezneciler, 34314 Istanbul, Turkey; yesimakbaba34@gmail.com
2   Department of Material Science and Engineering, National Cheng Kung University, Tainan City 701, Taiwan; kaneko.s.al@m.titech.ac.jp
3   Kanagawa Institute of Industrial Science and Technology (KISTEC), Atsugi 243-0292, Japan
4   Tokyo Institute of Technology, Yokohama 226-8502, Japan
*   Correspondence: musa.can@istanbul.edu.tr; Tel.: +90-(533)-929-0718

**Abstract:** The study was focused on optimizing the procedure of synthesizing iron gallate (FeGa₂O₄) nanoparticles by mechanochemical techniques. Due to a lack of information in the literature about the sequence of synthesis procedures of FeGa₂O₄ structures, the study is based on the establishment of a recipe for FeGa₂O₄ synthesis using mechanochemical techniques. Rotation speed, grinding media, and milling durations were the optimized parameters. At the end of each step, the structure of the resulting samples was investigated using the X-ray diffraction (XRD) patterns of samples. At the end of the processes, the XRD patterns of the samples milled under an air atmosphere were coherent with the XRD pattern of the FeGa₂O₄ structure. XRD patterns were analyzed employing Rietveld refinements to determine lattice parameters under the assumption of an inverse spinel crystal formation. Furthermore, a fluctuation at band gap values in the range of 2.39 to 2.55 eV was realized and associated with the excess Fe atoms in the lattice, which settled as defects in the crystal structures.

**Keywords:** spinel oxide semiconductors; nanoparticle catalysts; inverse spinel; mechanochemical synthesis; gallates

## 1. Introduction

Spinel oxide semiconductors are widely investigated materials due to their attractive performance in a variety of technological applications, such as photocatalytic fuel cells, biomaterials, and sensors [1–5]. The chemical formula of spinel oxides can be defined as a cubic structure with a formula of AB₂O₄, where A and B represent a divalent metallic cation at the tetrahedral site and tetravalent metallic cations at the octahedral sites, respectively [6,7].

Among the vast variety of spinel oxide semiconductors, iron gallate (FeGa₂O₄) has recently been considered an outstanding material due to its magnetic properties; piezoelectric, magnetoelectric, and magneto-optical performance; and cathodoluminescent features [8–12]. FeGa₂O₄ has a cubic spinel crystal structure with a space group of Fd-3m and a lattice parameter of a = 8.385 Å [13].

FeGa₂O₄ was firstly investigated due to its piezoelectric and magnetoelectric properties. Nowadays, FeGa₂O₄ structures show remarkable properties in Li-ion capacitors as anode materials or in biotechnological applications [14–16]. The technological phenomena of structures composed of FeGa₂O₄ mainly originate from its inverse or normal spinel structures [17,18]. Studies have indicated that FeGa₂O₄ can be formed as either a full inverse structure or a partial inverse structure. However, studies on FeGa₂O₄ structures still need further investigation since the synthesis of FeGa₂O₄ structures is quite difficult. In the literature, the number of studies on a systematic method for the synthesis of FeGa₂O₄

structures are very few, and the processes are not easy to follow [10,11,13,14,18–21]. This is why there is still not enough information on synthesis processes.

$FeGa_2O_4$ structures also seem very attractive for use in thick/thin films. $FeGa_2O_4$ oxide thin films seem to be suitable materials for photocathodes due to their physical properties such as transparency to visible light, p-type electrical conductivity, high enough chemical resistivity to an acidic environment, and tunable band edge by sunlight to the redox potential of water. According to the needs of photocatalytic materials, $FeGa_2O_4$ structures appear suitable for photocatalytic fuel cell applications either as a thin film or a sponge-like thick film with nanowalls [22,23].

Mechanochemical syntheses are widely operated techniques used to produce a variety of functional materials [24]. Charge ratio, rotation speed, milling environment, and duration are parameters that may manage structural formation [25]. The synthesis parameters of the mechanochemical techniques were adjusted to synthesize $FeGa_2O_4$ structures, and the crystal structure of the synthesized particles was investigated by employing Rietveld refinements.

## 2. Experimental

Mechanochemical techniques were employed to synthesize iron gallate ($FeGa_2O_4$) nanopowders. During the synthesis processes, the parameters of rotation speed and milling environments were optimized. At each process, the same amount of Fe (purity 99.9% Sigma Aldrich), $Ga_2O_3$ (purity 99.9% Sigma Aldrich), and deionized water ($H_2O$) was used as the starting materials, and an excessive amount of $H_2O$ was added to the container as a lubricant material.

Each milling process was performed employing a Planetary Mono Mill pulverisette 6 (FRITSCH GmbH, Idar-Oberstein, Germany) inside of a stainless steel vial with 10 mm diameter stainless steel balls. Three separate procedures were performed. The samples were named as *FeGaO-01*, *FeGaO-02,* and *FeGaO-03*. The *FeGaO-01* sample was obtained by the rotation speed of 300 rpm. After every 12 h of milling, the container was opened to the atmosphere and again milled for another 12 h. At the end of 84 h of the milling procedure, $FeGa_2O_4$ particles were produced. For the *FeGaO-02* sample, the rotation speed was increased up to 360 rpm and, again, the vial was opened after every 12 h of milling. At the end of 36 h of milling, $FeGa_2O_4$ particles were produced and named *FeGaO-02*. For the final sample, without opening the chamber to the atmosphere, a continuous milling procedure was performed for 36 h at a rotation speed of 360 rpm to understand the necessity of opening the vial to the atmosphere after each 12 h of milling. The produced sample was named *FeGaO-03*. The procedures are summarized in Table 1.

**Table 1.** The mechanochemical parameters during the synthesis of the particles.

| Sample Name | The Rotation Speed (Rpm) | The Milling Environments | Times of Open to Ambient Atmosphere | Charge Ratio (Powder to Ball Weight Ratio) | Duration (Hour) |
|---|---|---|---|---|---|
| *FeGaO-01* | 300 | Air | 8 | 1:30 | 84 |
| *FeGaO-02* | 360 | Air | 3 | 1:30 | 36 |
| *FeGaO-03* | 360 | released gas * | 0 | 1:30 | 36 |

* indicates released gas during the milling process.

To determine the crystal structures, XRD patterns were recorded by a MiniFlex model X-ray powder diffractometer (XRD) produced by Rigaku Corporation, with Cu $K_\alpha$ radiation (1.5406 Å) in 2 θ range from 10 to 90°. The XRD patterns were obtained by a scan rate of 0.008°/min steps, which are suitable for Reitveld refinements. The FOOLPROOF and MATCH subprograms were utilized to obtain the structural parameters.

The bandgap calculations of each powder were investigated by a Shimadzu UV-2600 model UV-VIS spectrophotometer (Shimadzu Scientific Instruments, Kyoto, Japan). The UV-Vis spectra were taken in the wavelength range of 195 nm to 1400 nm. The bandgap

calculations for powder samples were performed by the data of diffuse reflectance spectra (DRS). Then, the DRS data were analyzed by using the Kubelka–Munk transformation, employing the UvProbe 2.70 subprogram.

## 3. Results and Discussion

The procedures were mainly based on synthesizing $FeGa_2O_4$ particles by employing mechanochemical techniques, without annealing steps. In the beginning, the starting materials were put into the vial and the particles were milled for 12 h periods at a 300 rpm rotation speed. At the end of milling periods, the vial was opened to the atmosphere to release the $H_2$ gasses. The XRD pattern of the product was taken, as shown in Figure 1a. At the end of each milling period until the end of the completed 84 h milling process, the powders contained more than one compound. According to the JCPDS pdf cards, we assumed that the possible compounds in our samples were $FeGa_2O_4$, $\alpha$- FeOOH, $\gamma$- FeOOH, $Ga_2O_3$, and $Fe_3O_4$. The peak positions of contamination compounds were identified by arrows on figures. At the end of 84 h of milling at 300 rpm, it was noticed that the XRD pattern of final powders was coherent with the $FeGa_2O_4$ diffraction pattern with the pdf card number of JCPDS 01-074-2229. On the figures, $FeGa_2O_4$ peak positions were identified and indexed. No contamination compound or element was detected on the final pattern. The final product was named *FeGaO-01*, as seen in Figure 1a.

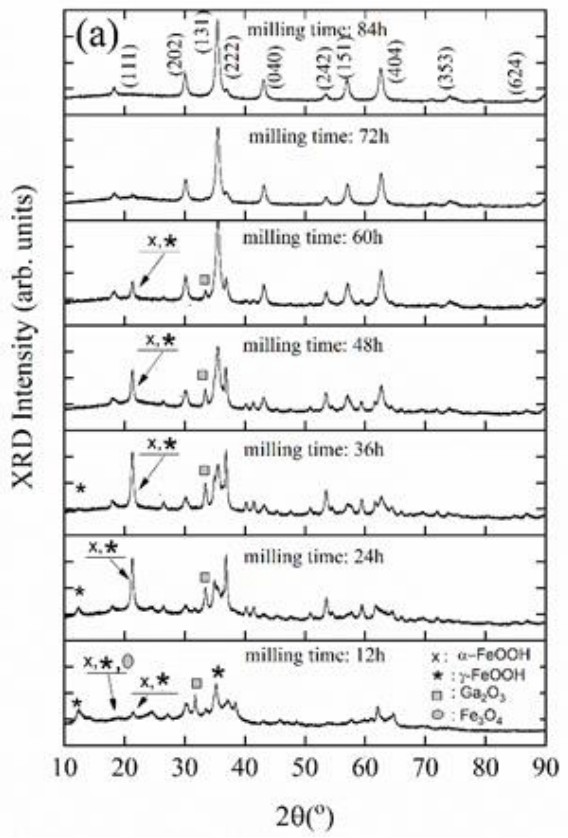 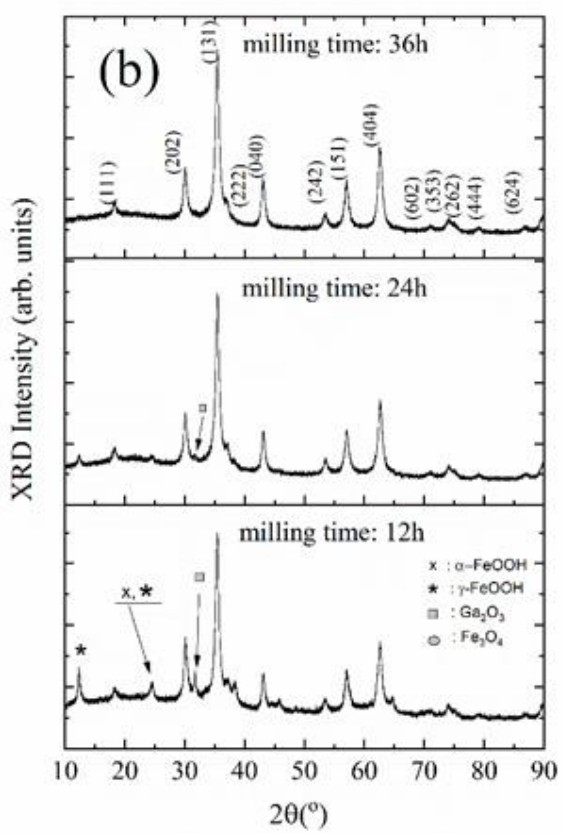

**Figure 1.** *Cont.*

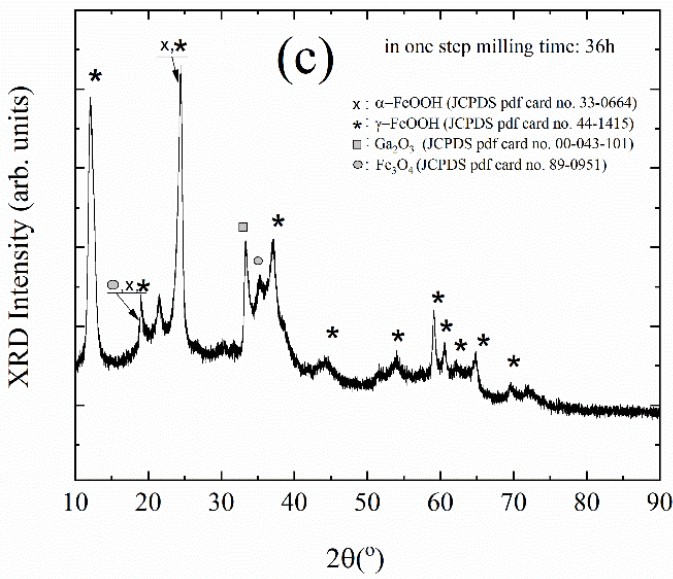

**Figure 1.** X-ray powder diffraction pattern of prepared samples by (**a**) a rotation speed of 300 rpm with 12 h milling periods, (**b**) a rotation speed of 360 rpm with 12 h milling periods, and (**c**) a rotation speed of 360 rpm without break in the milling. The diffraction patterns were indicated by the Joint Committee on Powder Diffraction Standards (JCPDS) powder diffraction file (pdf) card numbers, such as $FeGa_2O_4$, $\alpha$—FeOOH, $\gamma$—FeOOH, $Ga_2O_3$, and $Fe_3O_4$, which were identified by 01-074-2229, 33-0664, 44-1415, 00-043-101, and 89-0951, respectively).

To decrease the milling time, the milling parameters were optimized and the *FeGaO-02* sample was obtained. For *FeGaO-02*, the rotation speed was increased to 360 rpm, and the milling time intervals were kept at 12 h. The XRD patterns of samples, produced at each step, are shown in Figure 1b. As seen from XRD patterns, repeating the milling process three times was enough to produce $FeGa_2O_4$ nanoparticles. Then, the process was performed at a 360 rpm rotation speed non-stop for 36 h, without a break in order to understand the need for taking a break. The sample produced without taking a break to release the $H_2$ gases out of vial was named *FeGaO-03*. The XRD pattern of *FeGaO-03* is shown in Figure 1c. As seen in Figure 1c, the milling without a break was not enough to synthesize $FeGa_2O_4$ nanopowders. The expected chemical reaction during the milling process can be shown by the chemical Equation (1):

$$Fe + Ga_2O_3 + H_2O \rightarrow FeGa_2O_4 + H_2 \tag{1}$$

As seen in the chemical reaction, $H_2$ gasses were expected to release into the vial. $H_2$ gasses can prevent the reaction from continuing, as observed for $Fe_3O_4$ particles that in situ produced $H_2$ and hindered the formation of a new crystal phase during milling [26]. During the milling process, opening the vial releases the $H_2$ gasses, which exit the vial and allow the formation of the chemical reaction under the ambient atmosphere to finalize as $FeGa_2O_4$.

During the milling processes, we expected a large strain to form in the powders. Broadening at the XRD peaks was expected as the indicator of the strain formed in the particles. That is why, for the powders, XRD peaks were associated with particle size distribution and the strain with the powders [27–31]. In general, the average particle size was calculated using Debye–Scherrer's formula, as shown in Equation (2), and the strain originating from crystal imperfection and distortion can be calculated by Equation (3). To obtain the average particle size distribution and understand the strain effects, a general formula was defined as the uniform deformation model (UDM). The UDM was defined

by the equation shown in (4), which provides the effect of intrinsic strain associated with crystal size [28–31].

$$L = \frac{K\lambda}{\beta_{hkl}\cos\theta} \tag{2}$$

$$\varepsilon = \frac{\beta_{hkl}}{4\tan\theta} \tag{3}$$

$$\beta_{hkl}\cos\theta = \frac{K\lambda}{L} + 4\varepsilon\sin\theta \tag{4}$$

In the equation, the $\beta_{hkl}$, $L$, $K$, $\lambda$, $\varepsilon$, and $\theta$ symbols indicate the full width at the half-maximum of the diffracted peak, indicating the broadening of a peak, the average particle sizes, a constant defined as the shape factor (=0.9), the wavelength of the CuK$\alpha$ radiation (1.5406 Å), the strain, and the angle of diffraction, respectively.

From broadening XRD peaks, the particle sizes and strain associated with the samples of *FeGaO-01* and *FeGaO-02* were calculated by Equation (4), as shown in Figure 2a,b, respectively. The average particle size calculations were obtained from the diffraction peaks of (111), (202), (131), and (040). The crystallite sizes for the *FeGaO-01* and *FeGaO-02* samples were found as 14.4 nm and 12.0 nm, respectively. Furthermore, the microstrain of the particles was obtained by employing Equation (3), and the data are shown in Table 2. The microstrain values of the *FeGaO-01* and *FeGaO-02* samples, shown in Figure 2c,d, were calculated as $11.06 \pm 06 \times 10^{-2}$ and $1.1 \pm 0.2 \times 10^{-3}$, respectively. The microstrain is associated with a lattice dislocation density [32]. The longer time milling causes the formation of a 10 times higher microstrain with particles.

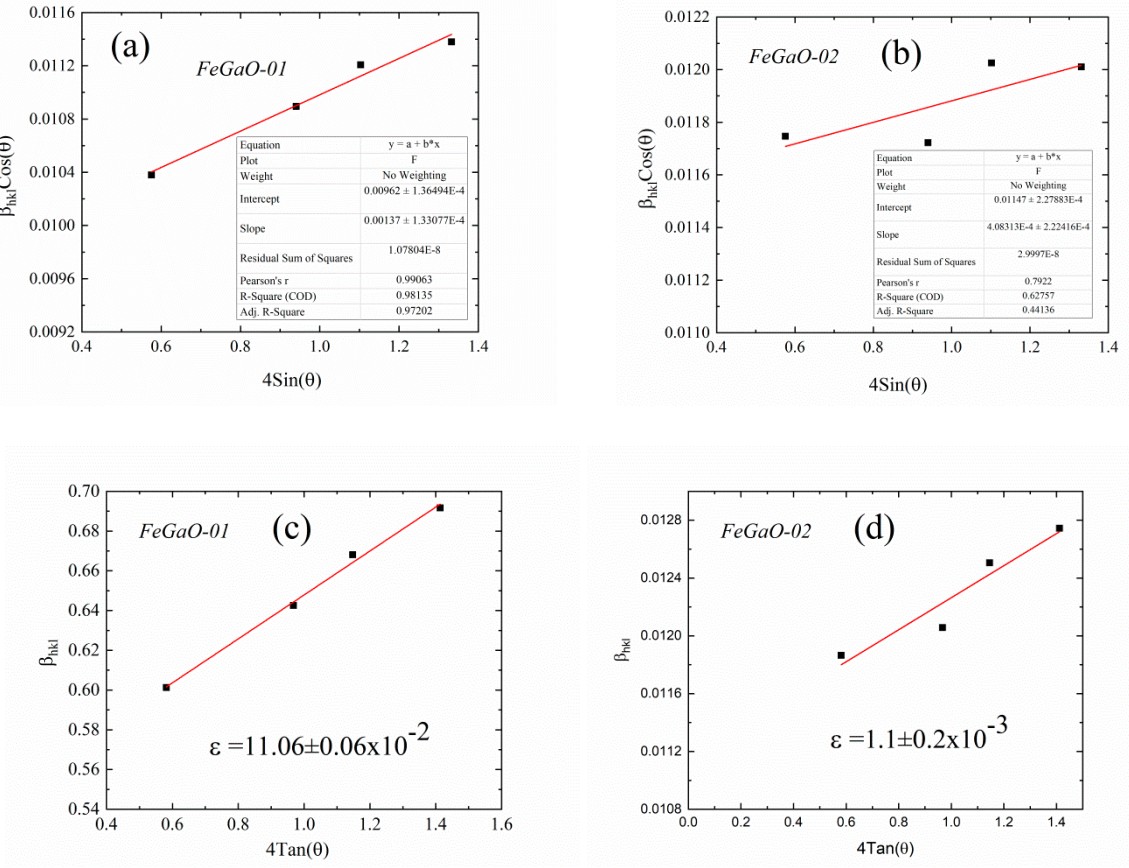

**Figure 2.** Fits to the uniform deformation model (UDM) for the samples to calculate the particle sizes of (**a**) *FeGaO-01* and (**b**) *FeGaO-02*, and to calculate the microstrain with particles of (**c**) *FeGaO-01* and (**d**) *FeGaO-02*.

**Table 2.** Rietveld refinement data and the band gap values of samples.

| Sample Name | Particle Size (nm) | Microstrain of Particles | Rietveld Refinement | | | | | | | | | | Band Gap (eV) |
| | | | Space Group | Bragg R Factors ($R_B$) | $\chi^2$ Values | The Lattice Constant a (Å) | Occupancy Ratios | | | | | | |
| | | | | | | | Tetrahedral Site | | | Octahedral Site | | | |
| | | | | | | | Fe1 | Ga1 | O3 | Fe2 | Ga2 | | |
| *FeGaO-01* | 14.4 | 11.06 $\pm 0.06 \times 10^{-2}$ | F4d₁32 (No. 210) | 0.842 | 0.368 | 8.3687 $\pm 0.0001$ | 0.0693 | 0.1540 | 0.4417 | 0.1540 | 0.0693 | 2.39 $\pm 0.02$ |
| *FeGaO-02* | 12.0 | 1.1 $\pm 0.2 \times 10^{-3}$ | | 0.910 | 0.338 | 8.3790 $\pm 0.0003$ | 0.0794 | 0.1397 | 0.4310 | 0.1760 | 0.0767 | 2.55 $\pm 0.01$ |

Then, Rietveld refinements employing the FullProf subprogram were performed to calculate the crystal parameters of the *FeGaO-01* and *FeGaO-02* samples. The crystal structure was assumed as the inverse spinel crystal formation. The inverse spinel crystal has the same XRD pattern of the $FeGa_2O_4$ spinel crystal. Under the assumption of the inverse spinel crystal, the crystal structure of $FeGa_2O_4$ was taken into account as a cubic and the cations, $Fe^{2+}$ and $Ga^{3+}$, were placed both the tetrahedral (0, 0, 0) and octahedral $(\frac{5}{8}, \frac{5}{8}, \frac{5}{8})$ sites. The space group of samples was assumed to be the F4d₁23 (No.210) space group, according to the fit of the MATCH subprogram. Then obtained parameters from the MATCH program were used in the FullProf program for structural analyses. The FullProf calculations are illustrated in Figure 3 and the data are shown in Table 2.

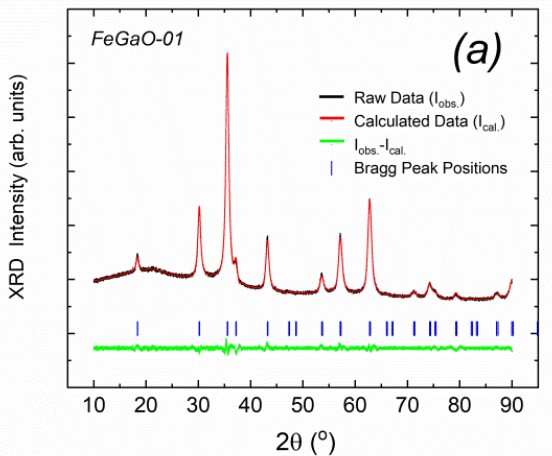 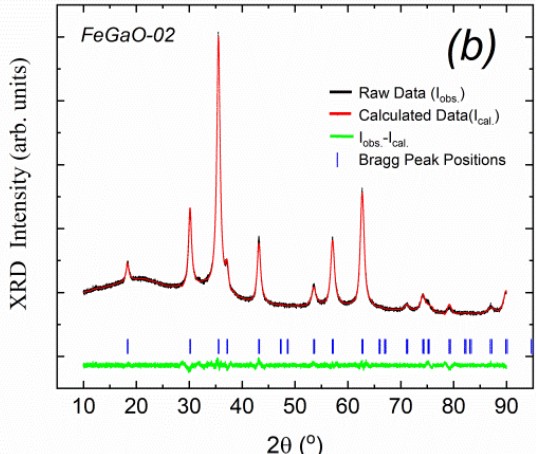

**Figure 3.** Rietveld refinements fits for the synthesized samples of (**a**) *FeGaO-01* and (**b**) *FeGaO-02*.

The bandgap values of each spectrum were analyzed from the reflectance spectra and each spectrum was modified by Tauc's Relation, as set out in Equation (5) [33]:

$$(\alpha h \upsilon )^n = A(h\upsilon - Eg) \tag{5}$$

where Eg, *h*, *υ*, *α*, A, and n represent the bandgap energy, Planck constant, frequency of light, absorption co-efficient, constant, and a constant for the direct band gap, 2, respectively. The fits are shown in Figure 4a,b, in which the horizontal axes are in the energy unit calculated by Equation (6):

$$E = h\upsilon = \frac{hc}{\lambda} \tag{6}$$

where *E*, *λ*, and *c* indicate the energy, wavelength, and speed of the light, respectively. The bandgap values of *FeGaO-01* and *FeGaO-02* were obtained as 2.39 ± 0.02 eV and 2.55 ± 0.01 eV. The difference between the calculated band gap values were assigned the

shallow defect states due to the formation of partial inverse spinel states, or to excess Fe atoms, which could have come from the miller vial during the milling processes [34].

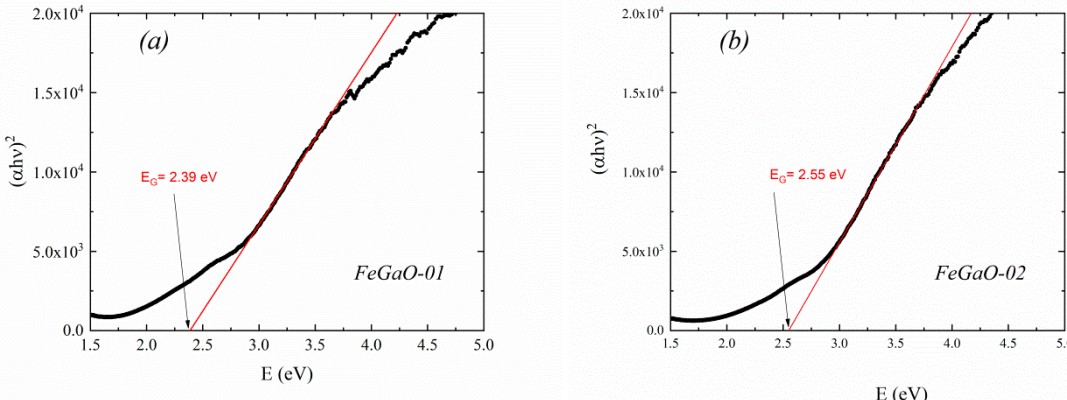

**Figure 4.** UV-vis spectra of the (**a**) *FeGaO-01* and (**b**) *FeGaO-02* samples.

As observed in the literature, excess atoms can stay in the crystal lattice as interstitial defect states [35]. These defect states create either donor or acceptor levels, which have electronic energy levels very close to band gap edges, possibly narrowing the band gap values [36].

## 4. Conclusions

The research was intended to optimize mechanochemical techniques to synthesize $FeGa_2O_4$ particles. The XRD patterns revealed that the $FeGa_2O_4$ particles were synthesized without any contamination elements or compounds. Replacement atoms in the tetrahedral and octahedral sites were analyzed by employing Reitveld refinement. Calculations were performed by assuming that the crystal configurations of the particles were formed as full or partial inverse spinel crystal structures. The bandgap values of $FeGa_2O_4$ were calculated as $2.39 \pm 0.02$ eV and $2.55 \pm 0.01$ eV. Measuring the two different bandgap values indicated the formation of shallow energy levels originating from defect energy states, which were close to either conduction or valance energy levels. Furthermore, the 84 h of milling caused the formation of a 10 times higher microstrain, according to 36 h milled particles. The high microstrain value broadened the peak values, and thus the result of the high microstrain was the decreased band gap value of the particles.

**Author Contributions:** Conceptualization, M.M.C.; methodology, M.M.C.; formal analysis, M.M.C. and Y.A.; investigation, M.M.C. and Y.A.; data curation, M.M.C. and Y.A.; writing—original draft preparation, M.M.C.; writing—review and editing, M.M.C. and S.K.; supervision, M.M.C.; project administration, M.M.C.; funding acquisition, M.M.C. All authors have read and agreed to the published version of the manuscript.

**Funding:** This research was funded by [the Scientific and Technological Research Council of Turkey (TÜBİTAK)] grant number [118F373]. This research was also funded by [the Scientific Research Projects Coordination Unit of Istanbul University], grant number [FYL-2021-38050].

**Institutional Review Board Statement:** Not applicable.

**Informed Consent Statement:** Not applicable.

**Data Availability Statement:** The data about space groups was acquired from the MATCH subprogram, and they have not given their permission for researchers to share their data. Data requests can be made to the company CRYSTAL IMPACT via this email: info@crystalimpact.de.

**Conflicts of Interest:** The authors declare that they have no conflict of interest.

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
