# Peer review of "Synthesis of Iron Gallate (FeGa2O4) Nanoparticles by Mechanochemical Method"

_coatings, doi:10.3390/coatings12040423_

Round 1

Reviewer 1 Report

X-Ray is capitalised. You have used Cu radiation to characterise a sample containing Iron. Cu causes fluorescence and there are absorption issues which you have neither accounted for or even mentioned in your methodology. This needs to be addressed otherwise your XRD results are in doubt. For example yes to some degree the detector can filter the fluorescence but that also means that minor peaks will not be detected. In your resubmission please don't just quote this sentence but actually take it into account with your analysis. You need to give the manufacturer of the diffractometer and Rietveld software. Rietveld results where were the errors in the occupancy, I seriously doubt that they are exactly the same for two different samples. They are sites not sides. For crystallites of this size the equation you have used has issues please refer to https://www.mdpi.com/2079-4991/10/9/1627/pdf line 49 should be on the synthesis line 95-96 does not make sense and needs to be rewritten Several other grammar issues, the manuscript would benefit from an online grammar checking program. line 103-104 should read placed on both sites

Reviewer 2 Report

In this paper author described the synthesis process of FeGa2O4 and analysed the structural properties. This paper needs a minor revision before publishing in Coatings journal. My comments are given below.

  1. Author should discuss Table 1 in more detail
  2. Fig1c missing.
  3. It is desirable to include microstructures of FeGa2O4.

Reviewer 3 Report

The presented article is devoted to the investigation of iron gallate synthesized by the ball milling. This article is very short and contains no any discussion. The quality of the presentation, including English, leaves much to be desired. I think that this article is not suitable for publication. Some other comments follow below.
1) In the experimental section, it is necessary to indicate the equipment for mechanochemical processing, the ball-to-powder ratio, the material of the balls and their dimensions.
2) Caption to fig. 1 should be presented in detail for figures a, b and c. In addition, fig. 1c is missing and needs to be added.
3) The reason that milling without interruption was not enough for the synthesis of FeGa2O4 nanopowders needs to be explained.
4) The calculation of the crystallite size (not the particle size) was carried out incorrectly, since the Scherrer formula does not take into account the microstrain introduced by the ball milling. In addition, the calculation from different diffraction peaks should give different sizes.

Reviewer 4 Report

  1. In the paper, it is mentioned that the XRD pattern of FeGaO-03 was shown in Figure 1c. However, there is no Figure 1c in the paper.
  2. The band gap values of FeGaO-01 and FeGaO-02 were obtained as 2.390.02eV and 2.550.01eV. The difference between calculated band gap values were assigned the shallow defect states due to formation of partial inverse spinel states or excess Fe atoms, coming from miller vial during milling processes. More detailed explanations or discussion are needed. For example, which case could have the shallow defect states and which case could have excess Fe atoms.
  3. Regarding material properties, only band gap values were determined. The band gap is a major factor determining the electrical conductivity of a solid. No specific optical properties of iron gallate (FeGa2O4) are provided in this paper. Authors are suggested to explain this issue or to measure their optical properties.
  4. Authors are suggested to discuss how to apply the as-synthesized powders to form a thin film or coating. 

Reviewer 5 Report

Submitted by authors paper entitled 'Structural and Optical Analyses of FeGa2O4 Particles Synthesized via Mechanochemical Route' is properly written. As the authors mentioned in the introduction, the synthesis of iron gallate is very complex and not easy to follow. Thusly, the submitted paper should be very interested to readers and shows valuable results. However, some errors have to be improved:

  1. In Tab. 1., the charge ratio is missing
  2. Fig. 1c is missing
  3. The manufacturer of the used diffractometer and spectrophotometer should be added.
  4. Line 106-107: 'In addition, the lattice constant, a, and occupancy ratios of each atom were illustrated in Table 1 and Table 2.' There are no structural data in table 1. Please change.
  5. For calculation of crystallite size, the Rietveld refinement instead of the Scherrer equation should be used. Rietveld refinement allows to calculate of lattice parameters, crystallite size, and lattice strain simultaneously, thus it is a better tool, due to considering also lattice strain. Why authors used the Scherrer equation instead of Rietveld refinement?

Round 2

Reviewer 1 Report

Thank you for making the suggested corrections. Some of the English could still be improved as that will help the reader understand your paper.

Author Response

Thank you for your comments which improved the study.

Reviewer 3 Report

After revision, some comments still remain.
1) The equipment for milling needs to be specified.
2) The charge ratio should be rechecked: 1:30 or 30:1?
3) The caption of the figure 1 should indicate what figures a, b and c refer to.
4) In order to develop the discussion, it is desirable to focus on the process of formation of FeGa2O4 phase. So, it can be seen from Fig.1a that the intensity of the Fe3O4 phase at an angle of slightly more than 20 deg sharply increases after 24 h and then gradually decreases. This fact looks strange and needs an explanation.
5) Intermediate reactions, including those with the formation of Fe3O4, should also be indicated.
6) In the response, the authors mentioned some SEM micrographs of particles that they added to the article. Unfortunately, I didn't find them.
7) The specified accuracy of 0.01 nm in determining the crystallite  sizes of cannot be achieved, so their values should be rounded to the nearest whole number. In addition, please provide the calculated values for microstrain using equation (3).
8) The text inside the frame in fig. 2 are not readable.

Author Response

Thanlk you for your review. Please see the attachment.

Reviewer 4 Report

Authors have replied properly the comments made by reviewers. 

Author Response

(The authors gave the same response as above.)

Reviewer 5 Report

Authors strongly improved their manuscript and after minor text editing (fonts, fonts size, etc.), it will be sufficient for publication in Coatings. Please also double-check for typos.

Author Response

(The authors gave the same response as above.)
